# Fecal Microbiota Dynamics Reveal the Feasibility of Early Weaning of Yak Calves under Conventional Grazing System

**DOI:** 10.3390/biology11010031

**Published:** 2021-12-26

**Authors:** Jianbo Zhang, Peng Wang, Renqing Dingkao, Mei Du, Anum Ali Ahmad, Zeyi Liang, Juanshan Zheng, Jiahao Shen, Ping Yan, Xuezhi Ding

**Affiliations:** 1Key Laboratory of Yak Breeding Engineering, Lanzhou Institute of Husbandry and Pharmaceutical Sciences, Chinese Academy of Agricultural Sciences, Lanzhou 730050, China; zhangjb9122@163.com (J.Z.); dumeicaas@126.com (M.D.); anum2017@lzu.edu.cn (A.A.A.); liangzeyi1@163.com (Z.L.); sfazjs1228@sina.com (J.Z.); shenjiahao@163.com (J.S.); pingyanlz@163.com (P.Y.); 2Key Laboratory of Veterinary Pharmaceutical Development, Ministry of Agricultural and Rural Affairs, Lanzhou Institute of Husbandry and Pharmaceutical Sciences, Chinese Academy of Agricultural Sciences, Lanzhou 730050, China; m18747115215@163.com; 3Gannan Institute of Animal Husbandry Science, Hezuo 747000, China; dingkao2020@163.com

**Keywords:** yak, cattle, early weaning, fecal microbiota, core microbiota

## Abstract

**Simple Summary:**

Yak (*Bos grunniens*) is the most economically and culturally important domestic bovine species adapted to the extreme ecological environment of the Qinghai–Tibetan Plateau (QTP), which provides milk, meat, transportation, fuel (yak dung), and wool for local nomads as well as major sources of income. Calves are an important part of the sustainable development of the yak industry on the QTP, and the quality of calf rearing directly determines the production performance of adult animals. Under the traditional grazing management, late weaning (>180 days) of yak calves seriously affects the improvement of their production performance. A comparative study of fecal microbiota dynamics of yak and cattle (*Bos taurus*) calves in different months after weaning will help to understand the changes in intestinal microbiota structure, and will aid in in improving growth rate and survivability of early weaned calves. Our research will contribute to the development of appropriate strategies to regulate the gut microbiome and thus improve the growth and health of the grazing ruminants on the QTP.

**Abstract:**

Background: The gut microbiota plays an important role in the health and production of animals. However, little information is available on the dynamic variations and comparison of intestinal microbiota in post-weaning yak calves living on the QTP. Methods: We explored the fecal bacterial microbiota succession of yak calves at different months after early weaning (60 d) compared with cattle calves by 16S rRNA gene amplicon sequencing and functional composition prediction. Results: We found no significant difference in blood biochemical parameters related to glucose and lipid metabolism between yaks and calves in different months after weaning. The core fecal bacterial microbiota from both species of calves was dominated by *Ruminococcaceae*, *Rikenellaceae*, and *Bacteroidaceae*. The fecal microbial community has a great alteration within the time after weaning in both cattle and yak calves, but cattle showed a larger change. After five months, the microbiota achieves a stable and concentrated state. This is also similar to the functional profile. Conclusions: Based on the exploration of dynamic changes in the fecal microbiota at an early stage of life, our results illustrated that there were no negative effects of intestinal microbiota succession on yak calves when early weaning was employed.

## 1. Introduction

The Qinghai–Tibetan Plateau (QTP) offers extreme environments with hypoxia, high altitude, long cold season, and limited forage resources, making it suitable for investigating species radiation and high-altitude adaptation of organisms [1]. The yak, a herbivore species exclusively inhabiting the QTP and adjacent mountainous regions, evolutionarily diverged from cattle about 4.4 to 5.3 million years ago [2]. In long-term evolution, yaks developed a unique rumen microecological system with a strong fiber-degrading ability to resist the extreme environment and seasonal forage supply imbalance following synergistic selection [3]. To date, several studies have found that yak is superior to cattle due to feeding and grazing behavior [4], digestive organ structure [5,6], nitrogen use efficiency [7], low rumen methane emission [8], and interseason energy utilization efficiency [9,10]. Meanwhile, the abundance of uncultured rumen microbial species was higher in the naturally grazing yak compared with house-farmed cattle [11]. A recent study argued that the reason why yak adapted to harsh environments and long-term nutritional stress on the QTP is related to the enrichment of key genes for volatile fatty acid (VFA) fermentation pathways in the rumen microbiome, whereas the methanogenesis pathways were enriched in cattle [3]. Furthermore, the study described that the maturation trajectory of unique microbiota and the inter and intra-associations among bacteria, fungi archaea, and protozoa in the rumen of grazing yaks through their lifespan [12]. Most research has emphasized the rumen microbiota of yak; however, the development of gut microbiota in weaned yak is poorly understood.

The gut microbiota of mammals has been increasingly recognized as a key factor affecting the health, development, and productivity of animals [13,14]. Several studies have found that mutualistic relationships between the host and its symbiotic gut microbiota perform a crucial role in the host’s health, increase host resistance to pathogenic bacteria, and are critical to the development and maturation of the immune system [15,16]. Moreover, it has been also reported that the establishment of intestinal microbiota during early life is associated with calf health and growth (neonatal diarrhea, pneumonia, and weight gain), and colonization by enteric pathogens might be responsible for dysbiosis in the intestinal microbiota of neonatal diarrhea [17,18,19]. Some studies have shown that microbial fermentation in the hindgut may be responsible for up to 30% of cellulose and hemicellulose degradation in ruminants [20,21]. Importantly, ruminants have special microbiota composition that allows them to convert human inedible low-quality plant fiber into high-quality products (meat, milk) for human consumption [18]. Therefore, a better understanding of the succession of the gut microbiota of ruminants is crucial in optimizing their health and production efficiency.

The early colonization of the gut microbiota begins at birth in ruminants and continues through successive waves of colonization, then attaining a stable shape later in life [22,23]. The study described that the dynamic of gut microbiota before weaning exerts a lasting impact on the health of adult ruminants and their products [24,25]. The colonization of gut microbiota is a complex procedure impacted by the 2-way relationship between host and microbiota, as well as a range of external factors, such as maternal microbiota, diet, parturition, and antibiotics [18]. However, the gut microbiota of calves gradually turned to maturity and stability with the intake of solid feed during the weaning period [26]. Recent studies have found that weaning has a greater impact on the development of gastrointestinal microbiota in calves than weaning strategies. At present, it is not clear how the succession of gut microbial communities of yak and cattle calves’ changes during various periods after weaning.

In the early stage, our studies have found that the intestinal microbial sources and colonization of yak and cattle calves at various developmental stages before weaning, and found that maternal fecal microbiota was the main source of gut microbiota of calves. To further explore the dynamic changes of host metabolism and intestinal microbiota in weaned yak and cattle calves, fecal and blood samples were continuously collected from eight yak and cattle calves at different months after weaning, and the characteristics of fecal microbiota were examined by the 16S rRNA gene sequencing method. This study will provide the basic understanding of fecal microbiota in yak and cattle calves, expecting to find ways such as microbiota manipulation to enhance the health and growth of ruminants during the complete production cycle.

## 2. Materials and Methods

### 2.1. Animal and Sampling

All the animals involved in this study were from the same herd and they all grazed together in an alpine meadow on the QTP, where the average altitude was 3300 m and the average annual temperature was 4 °C. Initially, we selected a total of 20 pregnant animals—yak (*n* = 10) and cattle (*n* = 10)—but their exact gestation period was not known. Both yak and cattle calves were born naturally, fed with milk by maternal suckling, and grazed on the same native pasture (without any concentrate supplementation) of Yangnuo Specialized Yak Breeding Cooperative (34°43′19.66′′ N, 102°28′49.51′′ E) at Xiahe county of Gannan Tibetan Autonomous Prefecture, Gansu Province, China. All the animals grazed from 7 a.m. to 6 p.m., and the samples were collected before the morning grazing. All animals included in this study were healthy during our sampling period and received no recorded therapeutic or prophylactic antibiotic treatment. In addition, the calves were customarily weaned and managed separately by herders about two months after birth. In the early stage, we have assessed the fecal microbiota succession of yak and cattle calves at different weeks after birth as well as the modes of transmission of maternal symbiotic microbes to their calves’ intestinal microbiota colonization. We found that the maternal fecal microbiota was the main source of the intestinal microbiota of calves, and the intestinal microbiota of yak calves reached a relatively stable state earlier than that of cattle calves.

To further explore the dynamic changes of host metabolism and intestinal microbiota in weaned yak and cattle calves, fecal and blood samples were collected in different months after weaning. From September 2019 to May 2020, a total of 64 fecal samples were continuously collected from yak (YW, *n* = 8) and cattle calves (CW, *n* = 8) at 1 (1M), 2 (2M), 5 (5M), and 8 (8M) months after weaning by inserting a gloved finger into the anus of the calf to stimulate defecation (Figure 1). All samples (*n* = 64) were frozen immediately in the liquid nitrogen, taken to the laboratory, and stored at −80 °C until DNA extraction. Meanwhile, the blood samples (*n* = 55) were collected from each animal by puncture of the jugular vein into non-oxalate tubes and taken to the laboratory and stored at −20 °C.

### 2.2. Detection of Blood Biochemical Indexes

The blood was centrifuged at 3500 rpm (15 min at 4 °C) and the supernatant (serum) was collected and introduced into new tubes for the subsequent biochemical analyses of the concentrations of serum glucose (GLU), triacylglycerols (TG), total cholesterol (TC), low-density lipoprotein cholesterol (LDL-C), and high-density lipoprotein cholesterol (HDL-C) using the Mindray BS-240VET Automatic Hematology Analyzer (Mindray Corporation, Shenzhen, China).

### 2.3. DNA Extraction and Illumina Sequencing of 16S rRNA Genes

The total genomic DNA from all the samples (*n* = 64) was extracted by the hexadecyl trimethyl ammonium bromide (CTAB) method [27]. The concentration and purity of DNA were checked on 1 % agarose gels. DNA was diluted to a final concentration of 1 ng/μL using sterile distilled water. The bacterial V4 region of 16S rRNA gene was amplified using F515/R806 universal primers [28] under the following conditions: initial denaturation at 98 °C for 1 min, followed by 30 cycles of denaturation at 98 °C for 10 s, annealing at 50 °C for 30 s, and elongation at 72 °C for 30 s, and finished by a final extension at 72 °C for 5 min. Amplicons were purified with Qiagen Gel Extraction Kit (Qiagen, Germany). Sequencing libraries were generated using TruSeq^®^ DNA PCR-Free Sample Preparation Kit (Illumina, San Diego, CA, USA) following the manufacturer’s recommendations, and index codes were added. The library quality was assessed on the Qubit@ 2.0 Fluorometer (Thermo Scientific, Waltham, MA, USA) and Agilent Bioanalyzer 2100 system. At last, the library was sequenced on an Illumina NovaSeq PE250 platform and 250 bp paired-end reads were generated (Novogene, Tianjin, China).

### 2.4. Bioinformatics and Statistical Analysis

The paired-end reads were assigned to samples based on their unique barcode [29] and then data were imported to QIIME2 (version: 2020.8.0) pipline for further analysis [30]. Briefly, (i) primers were removed by “qiime cutadapt trim-paired” (--p-minimum-length 200); (ii) sequences were denoised using dada2 algorithm (“qiime dada2 denoise-paired”) to obtain feature sequences (amplicon sequence variant, ASVs) and table (--p-trim-left-f 15 --p-trim-left-r 20 --p-trunc-len-f 0 --p-trunc-len-r 0 --p-n-threads 6) [31], features with frequency less than 4 were removed; (iii) the sequences from SILVA database (release 132) [32] were extracted using specific primers for V4 region to train Naive Bayes classifier for taxonomy assignment using “qiime feature-classifier classify-sklearn”, ASVs assigned to mitochondria and chloroplast were excluded from feature table.

Alpha diversity of fecal microbiota was characterized by Chao1 and Shannon diversity indices using “qiime diversity alpha” command line. Statistical comparison of the alpha diversity indices between group levels was performed using the Wilcoxon rank-sum test and Kruskal–Wallis test. For beta-diversity, we used “qiime diversity beta” to obtain distance matrices and principal coordinate analysis (PCoA) was performed based on weighted and unweighted Unifrac distance by “qiime diversity pcoa” and “qiime emperor plot”. Permutational multivariate analysis of variance (PERMANOVA) test and analysis of similarities (ANOSIM) test was both applied to test the difference between communities by “qiime diversity beta-group-significance” [33]. Additionally, we picked the ASVs occurring in all groups as the core ASVs, then visualized them using UpSet plot [34]. The linear discriminant analysis (LDA) effect size (LEfSe) algorithm was used for differential analysis to identify biomarker taxa [35]. The microbial functional prediction was conducted by PICRUSt 2 (Phylogenetic Investigation of Communities by Reconstruction of Unobserved States) [36] according to the standard method [37]. The predicted functional contents were summarized at Kyoto Encyclopedia of Genes and Genomes (KEGG) pathway hierarchy levels 2 for interpretation and subsequent analysis. Wilcox test and Benjamini-Hochberg FDR correction were used for pairwise group analysis. *p*-values were adjusted with false discovery rate and the corrected *p*-values < 0.05 were regarded as statistically significant.

## 3. Results

### 3.1. Comparison of Blood Biochemical Indexes in Post-Weaning Calve

To evaluate the health level of weaned calves, the key blood biochemical indexes related to glucose and lipid metabolism of yak and cattle calves were measured in different months after weaning. There were no significant differences in blood biochemical indexes related to glucose and lipid metabolism in yak and cattle calves at different months after weaning (ANOVA, *p* > 0.05, Table 1), such as GLU, TG, TC, LDL-C, and HDL-C. However, we found a significantly higher blood HDL-C in cattle calves after weaning than in yak calves (*t* test, *p* = 0.03, Table 1), and showed no statistical differences in other blood biochemical indexes (*t* test, *p* > 0.05). We found that calves have fully adapted to the effects of weaning through the metabolic level of calves in different months after weaning.

### 3.2. Diversity of Fecal Bacterial Microbiota in Post-Weaning Calve

The rarefaction curves and species accumulation boxplot results (Appendix A) showed adequate sequencing depth. We obtained a total of 5,738,602 (mean ± sd: 89,666 ± 6843 per sample) high-quality 16S rRNA gene sequences, which were denoised to 9803 ASVs.

The Shannon indices of fecal microbiota in yak calves remained stable after weaning (Kruskal–Wallis test, *p* = 0.126, Figure 2A), whereas the Chao1 species richness indices also remained stable (Kruskal–Wallis test, *p* = 0.251, Figure 2B and Appendix A). These results showed that bacterial richness and species diversity of yak calves remain relatively steady after weaning. However, we found that the species richness and diversity of the fecal microbiota of cattle calves in the fifth month after weaning were significantly higher than in the other months (Kruskal–Wallis test, Shannon, *p* = 0.0001; Chao1, *p* = 0.003, Figure 2C,D). Additionally, the species richness and diversity of fecal microbes in yak calves were significantly higher than that in cattle calves at 2 months after weaning (Willcoxon test, Shannon, *p* = 0.046; Chao1, *p* = 0.009, Appendix A), but the diversity of fecal microbiota in cattle calves at 5 months after weaning was significantly higher than that in yak calves (Willcoxon test, Shannon, *p* = 0.027). Notably, our results suggest that the gut microbiota of weaned yak calves is more stable than that of cattle calves in the same grazing system on the QTP.

### 3.3. Comparison and Structure of Fecal Bacterial Communities of Weaned Calves

Principal coordinate analysis (PCoA) based on unweighted and weighted UniFrac distance was performed to compare the bacterial community structure in calves after weaning (Figure 3 and Appendix A). PCoA plot showed clear age-based separation of fecal bacterial microbiota between 1–2 months and 5–8 months after weaning (Figure 3A,B). We found significant differences in fecal microbial communities between yak and cattle calves in different months after weaning by PCoA using unweighted UniFrac distance (PERMANOVA test, *p* = 0.001, Figure 3A; ANOSIM, *p* = 0.003, Appendix A), but the difference disappeared with the maturation of gut microbiota in calves. From the view of taxa composition, a total of 33 bacterial phyla were identified, in which three dominated the bacterial microbiota (average cumulative abundance = 95.92%), including Firmicutes, Bacteroidetes, and Proteobacteria (Appendix A). At the family level, we found that *Ruminococcaceae*, *Rikenellaceae**, Bacteroidaceae*, and *Lachnospiraceae* were prevalent in the fecal samples of both yak and cattle calves in different months after weaning (Figure 3C,D). At the genus level, we found that *Bacteroides*, *Ruminococcaceae UCG-005*, *Rikenellaceae RC9 gut group*, and *Prevotellaceae UCG-004* were prevalent in the fecal samples of yak and cattle calves in different months after weaning (Appendix A). These data suggested that there was a divergence between yak and cattle in the early growing stage; subsequently, the bacterial community structure tended to have a similar profile.

### 3.4. Core Fecal Microbiota in Weaned Yak and Cattle Calves

We further explored the core bacterial community structure and found that there were 696 ASVs shared in the fecal bacterial communities in yak and cattle calves after weaning (Figure 4A). Among these shared ASVs, most of them were assigned to Firmicutes (38.89%) and Bacteroidetes (34.81%). The core family contained *Ruminococcaceae* (24.80%), *Rikenellaceae* (12.16%), *Bacteroidaceae* (9.83%), and *Lachnospiraceae* (6.02%) (Figure 4B). At the genus level, *Ruminococcaceae UCG-005* (10.43%), *Bacteroides* (9.83%), and *Rikenellaceae RC9 gut group* (6.92%) dominated in all samples. In yak calves, 884 ASVs were shared (Appendix A). The core family of these shared ASVs was mainly *Ruminococcaceae*, *Rikenellaceae*, *Bacteroidaceae*, *Lachnospiraceae*, and *Prevotellaceae* (Appendix A). Among the fecal samples in cattle calves, 870 ASVs were shared (Appendix A). At the fifth month after weaning, we found that yak and cattle calves had 1289 and 1387 different AVSs (Figure 4A), respectively. Although yak and cattle calves have a large number of similar intestinal microbiota, there are still many special intestinal microbiotas at different developmental stages.

### 3.5. Differential Taxa between Weaned Yak and Cattle Calves

To further determine the influence of early weaning on the intestinal microbiota of calves, linear discriminant analysis effect size (LEfSe) was performed to determine whether calf fecal microbial community structure changed with age (Figure 5). LEfSe results showed that *Bacteroidaceae*, *Lactobacillaceae*, *Lachnospiraceae*, and *Succinivibrionaceae* were more abundant in cattle calves at 1 month after weaning than in yak calves (Figure 5C), but the relative abundance of Euryarchaeota in yak calves at 8 months after weaning was higher than in cattle calves. For weaned yak calves, *Prevotellaceae UCG-010* and *Ruminococcaceae UCG-014* were dominant at 1 month, and *Rikenellaceae RC9 gut group* increased at 5 months; Euryarchaeota was dominant at 8 months after weaning (Figure 5A). In contrast, Proteobacteria was dominated in the fecal microbiota of cattle at 1 month after weaning, and Firmicutes quickly became the dominant of fecal microbiota in cattle at 5 months after weaning, but *Ruminococcaceae* was dominant in fecal microbiota in cattle at 8 months after weaning (Figure 5B). Additionally, we found that the intestinal microbiota structure tended to be similar between yak and cattle calves with increasing age.

### 3.6. Potential Function of the Microbial Community in Weaned Calves

Veen analysis showed that the microbial function predicted by PICRUST2 in weaned yak and cattle calves was similar, among which 5733 microbial functional genes were shared (Figure 6A). Similar to the microbial composition results (Figure 3), principal component analysis (PCA) also showed a clear age-based separation between 1–2 months and 5–8 months, whereas no obvious separation observed between yak and cattle calves (Figure 6B). From 1 to 2 months after weaning, the fecal microbial function of yak and cattle calves was mainly distributed in environmental adaptation, lipid metabolism, carbohydrate metabolism, and glycan biosynthesis and metabolism (Figure 6C and Appendix A). At 5 to 8 months, it was dominated by transcription, translation, replication and repair, and the immune system (Figure 6C and Appendix A). However, we found that the potential function of fecal microbes in cattle calves was significantly higher than that in yak at 5 months after weaning (Figure 6C and Appendix A), such as viral protein families (Wilcox test, *p* = 0.0019; FDR, *q* = 0.030) and signaling molecules and interaction (Wilcox test, *p* = 0.0047; FDR, *q* = 0.046). Notably, these results further explained the differences in the intestinal microbial community structure between yak and cattle calves in different months after weaning.

## 4. Discussion

Yak is a necessity for the local herdsmen living on the QTP. The milk and meat of yak could be used as a food source and the yak feces as fuel [38]. Calves are an important part of the durable development of the yak industry on QTP, and the quality of calf raising directly influences the productivity of adult animals [1,39]. Studies reported that the dynamic changes of early life gut microbiota of young ruminants exert a lasting impact on both adult ruminant health and animal products [24,25]. Further studies have found that variations in microbial composition were higher in younger than in adult ruminants, suggesting that the gut microbiota changed easier at early life stages than at later stages [18,40]. Indeed, dietary interventions on the rumen microbiota of young ruminants was successful in achieving fairly persistent and long-term results [41,42,43]. However, less data are present on the dynamic changes of early life gut microbiota of some ruminants living in extreme environments on the QTP. Previously, we have explored the source and colonization of intestinal microbes in yak and cattle calves before weaning living in the same pasture on the QTP, and found that maternal fecal microbiota might be an important source of intestinal microbiota in pre-weaning calves. A recent study found that intestinal microbiota was influenced by weaning but not by strategies of weaning [26]. Therefore, this study aimed to further explore the characteristic changes of their intestinal microbiota (as represented by the feces) at different time points after weaning.

After weaning of calves, the fecal microbial communities of yak and cattle calves were mainly composed of Firmicutes and Bacteroidetes, which was supported by the results of other studies [17,44,45]. Many studies reported that Firmicutes and Bacteroidetes are widely distributed in the gut microbiota of many mammals, indicating their ecological and functional significance in the digestive tract [26,29,46]. Firmicutes play a key role in degrading cellulose into volatile fatty acids that are utilized by the hosts [47,48,49]. Bacteroidetes can degrade carbohydrates and proteins, which is important to the development of the gastrointestinal immune system [47,48,49]. We found a high relative abundance of Firmicutes in the fecal microbiota of calves, but the relative abundance of Bacteroidetes decreased. As our samples were collected from September to May of the following year, the quality of herbage gradually decreased and the cellulosic substances in herbage gradually increased during this period, which may lead to changes in the intestinal microbiota composition of weaned calves. In addition, several studies have shown that Proteobacteria maintain the stability of the structure of gut microbiota, and this is a key indicator of a healthy gut in mammals (fewer proteobacteria means healthier hosts) [50,51]. We found that the relative abundance of Proteobacteria in the feces of calves decreased gradually after weaning in this work, which indicates that the intestinal microecosystem tends to be stable, mature, and healthy as grazing and calves growing up. We found that the fecal core microbial communities of calves were dominated by *Ruminococcaceae*, *Rikenellaceae*, *Bacteroidaceae*, and *Lachnospiraceae*. As the important components of Firmicutes, *Ruminococcaceae* and *Lachnospiraceae* members are butyrate-producing bacteria and provide energy to the host by digesting dietary fiber present in plant cell wall into short-chain fatty acids [52,53]. However, less information is available about the metabolic function of *Rikenellaceae*, but the previous study has hypothesized that it may be associated with degradation of primary or secondary carbohydrates [54]. Studies have shown that *Bacteroidaceae* can degrade different plant polysaccharides, but studies on humans revealed that they do not respond effectively to fiber supplementation [55]. Low fermentable fiber and high energy-high protein feed in diets was conducive to the bacterial growth from the *Bacteroidaceae* family [56]. A recent study showed that *Bacteroidaceae*, *Lachnospiraceae*, *Rikenellaceae* as well as *Ruminococcaceae* were associated with intestinal health [57]. *Bacteroides* members are generally considered for their ability to digest a broad range of plant cell-wall polysaccharides [55,58]. We found that *Bacteroides* were prevalent in the intestinal microbiota of yak and cattle calves in various months after weaning. In addition, we found that there were no significant differences in blood biochemical indexes related to carbohydrate and lipid metabolism between yak and cattle calves in different months after weaning. These results indicate that the intestinal microbiota of weaned calves gradually tended to be mature and stable, and gut microbiota gradually colonized. This might be the influence of the same abiotic environment factor (temperature, oxygen, water, etc.) and similar grazing method (diet structure).

The colonization of gut microbiota in young ruminants is a complex and dynamic process that is susceptible to two-way interactions between the host and the microbes as well as influenced by a variety of external factors [26,44,59,60,61,62,63,64,65], including maternal microbiota, the birth process, diet, antibiotics, and weaning. Compared with barn feeding, characteristics of the gut microbiota in young animals reared under natural grazing conditions might be more complex and diverse. Our previous research has shown that maternal fecal microbiota might be an important source of intestinal microbiota in pre-weaning calves. Compared with cattle calves, no significant differences were observed in the fecal microbial composition of yak calves between 5 and 9 weeks after birth, indicating that yak might adapt to its natural extreme environment to stabilize its gut microbiota composition. However, no significant differences were recorded in the fecal microbial community between yak and cattle calves at 9 weeks after birth. Similarly, differences were observed in fecal microbial community characteristics (abundance, diversity, composition, structure, and function) between yak and cattle calves at the first month after weaning; however, these differences gradually disappeared with the increase of calf age. This means that the effect of host genetic factors on the intestinal microbiota is gradually lower than environmental factors [22,23,66]. Additionally, diet is one of the key elements that influence the composition of gut microbiota. An early introduction to forage can promote the stability and healthy development of the gut microbiota of neonatal calves [22,40,67,68]. In this study, yak and cattle calves after birth living with their mothers in natural grazing conditions might have provided calves with more opportunities to adapt to forage early and promote the development of gut microbiota. It was discovered that calves raised in the presence of older companions visited the feeder more frequently and for longer periods, which was thought to be the result of social learning [67,69]. The gut microbiota of young ruminants undergoes several waves of colonization and community alterations before stabilizing later in life [23,39,70], suggesting that the gut microbiota in the early life is susceptible to environmental exposure and dietary changes. This was supported by a recent study on the dynamics of the rumen microbiota in yak from birth to adult age [12]. Furthermore, we found that the fecal microbiota structure of cattle calves at the fifth month after weaning was more diverse compared with yak calves. We found that this period falls in January, but it is the most extreme month on the QTP, with the lowest temperatures and the least forage resources [1]. Therefore, this further indicates that the yak calves can better adapt to the extreme natural environment of the QTP in the long-term evolution.

Gut microbes in early life are important for many aspects of animal including immune [71,72], metabolic [73], and neurobehavioral traits [74]. Microbial amino acid metabolism, carbohydrate metabolism, and energy metabolism are crucial in the intestines and provide energy to the host [75]. Based on the predicted metagenomes of fecal microbiota, the microbial functions of yak and cattle calves were similar, in which the gut microbial functions mainly focused on environmental adaptation, lipid metabolism, carbohydrate metabolism, and glycan biosynthesis and metabolism at 1 to 2 months, but transcription, translation, replication and repair, and immune system at 5 to 8 months after weaning. Our study showed that the intestinal microbiota of calves in the early stage was mainly concentrated in the glucose and lipid metabolic pathways. With the change of calf age and grazing months, the function of fecal microbiota gradually focused on the pathways related to transcription and translation. These results suggest that the gut microbiome adjusts in time for the better survival of the host. However, the detailed function of intestinal microbiota needs to be further determined by techniques such as metagenomics, macrotranscriptomics, and metabolomics.

## 5. Conclusions

In this study, we investigated the dynamic changes in the fecal microbiota of yak and cattle calves inhabiting the same natural pasture during the post-weaning period. There were no significant differences in blood biochemical indexes related to carbohydrate and lipid metabolism between yak and cattle calves in different months after weaning. However, significant differences were observed in fecal microbial community characteristics (abundance, diversity, composition, structure, and function) between yak and cattle calves at the fifth month after weaning, whereas at a later stage, the fecal microbial profiles were similar as they were grazing after weaning. Thus, based on the exploration of dynamic changes in the fecal microbiota at an early stage of life, our finding may be helpful to make rational strategies to manipulate the intestinal microbiota, thereby improving the health and growth of grazing ruminants through the complete conventional production processes in the region of the QTP.

## Figures and Tables

**Figure 1 biology-11-00031-f001:**
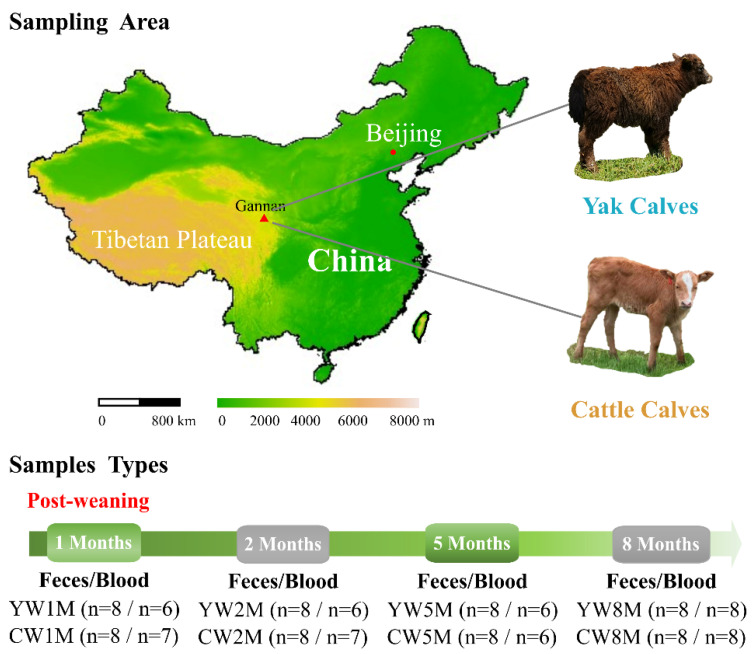
Experimental design and sample collection. Fecal and blood samples were continuously collected from yak and cattle calves in different months after weaning on the same pasture.

**Figure 2 biology-11-00031-f002:**
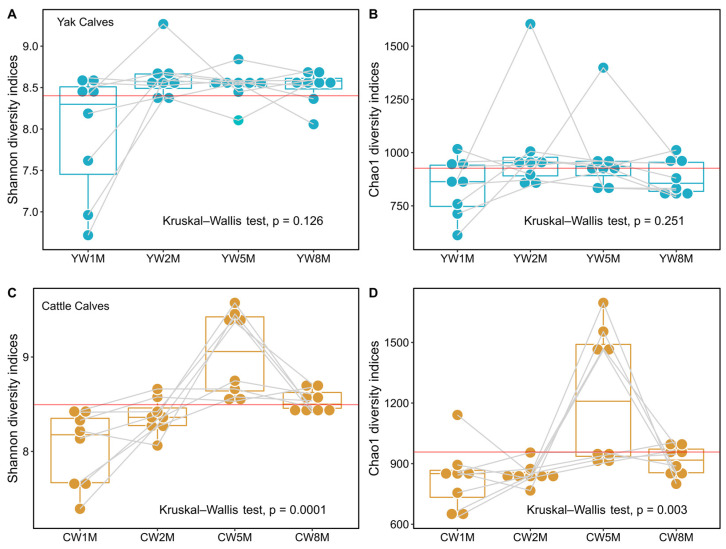
Alpha diversity of fecal microbiota of yak and cattle calves after weaning. The Shannon and Chao1 species richness indices in yak (**A**,**B**) and cattle (**C**,**D**) calves are shown by box plots. The Chao1 species richness and Shannon diversity indices of fecal microbes at different months after weaning were statistically analyzed by the Kruskal–Wallis test. The red line represents the intergroup average. The light gray line represents the diversity of fecal microbiota in the same individual in different months after weaning.

**Figure 3 biology-11-00031-f003:**
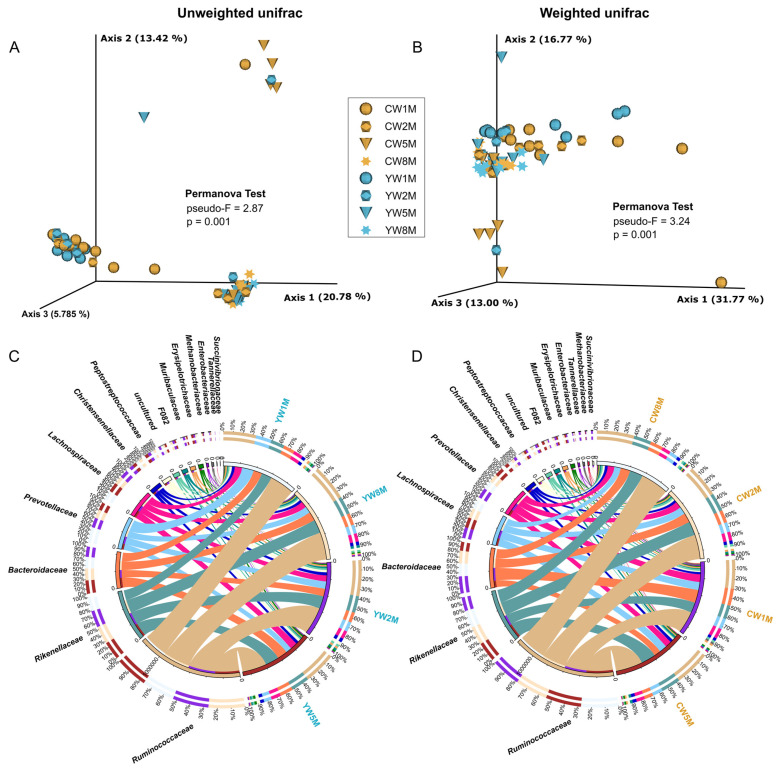
Beta diversity analysis of the fecal microbial community in yak and cattle calves at different months after weaning. (**A**) PCoA plot based on unweighted unifrac distance shows the differences between yak and cattle calves in different months after weaning. (**B**) PcoA plot based on weighted unifrac distance shows the differences between yak and cattle calves in different months after weaning. Circos diagram shows the composition of fecal microbiota at the family level (Top15) of yak (**C**) and cattle calves (**D**) in different months after weaning, respectively. The length of the bars on the outer ring and the numbers on the inner ring represent the percentage of relative abundance of genera detected in each sample and the number of sequences in each sample, respectively. The bands with different colors show the source of each sequence affiliated with different clusters.

**Figure 4 biology-11-00031-f004:**
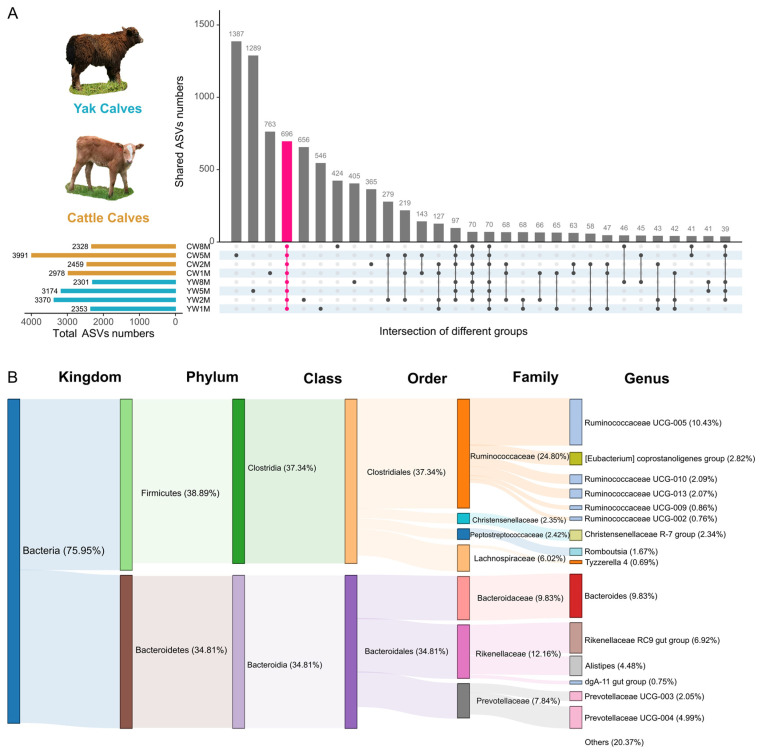
Analysis of the core fecal microbial community between yak and cattle calves in different months after weaning. (**A**) UpSet plots of common ASVs in fecal samples from yak and cattle calves in different months after weaning. The vertical bars or intersections represent the number of ASVs that were regulated by one or more samples type (intersecting conditions). The ASVs in each intersection were color-coded according to the meaning of their set. (**B**) Sankey diagram based on 696 shared ASVs at different taxonomic levels.

**Figure 5 biology-11-00031-f005:**
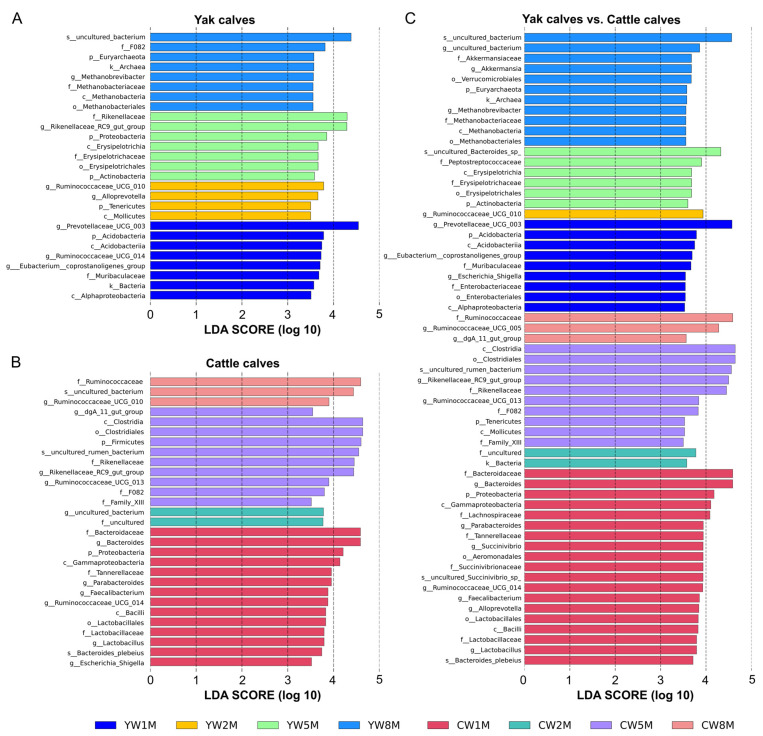
LEfSe analysis based on bacterial abundance data. LDA effect size (LDA score) comparison of fecal microbiota in yak calves (**A**), cattle calves (**B**), and between yak and cattle calves (**C**). The LDA cut-off score is 3.5. Letters in front of ASVs represent taxonomic levels (p, phylum; c, class; o, order; f, family; g, genus; s, species). The bar chart shows the different species in the fecal microbiota of calves in different months after weaning. Different colors represent different months after weaning.

**Figure 6 biology-11-00031-f006:**
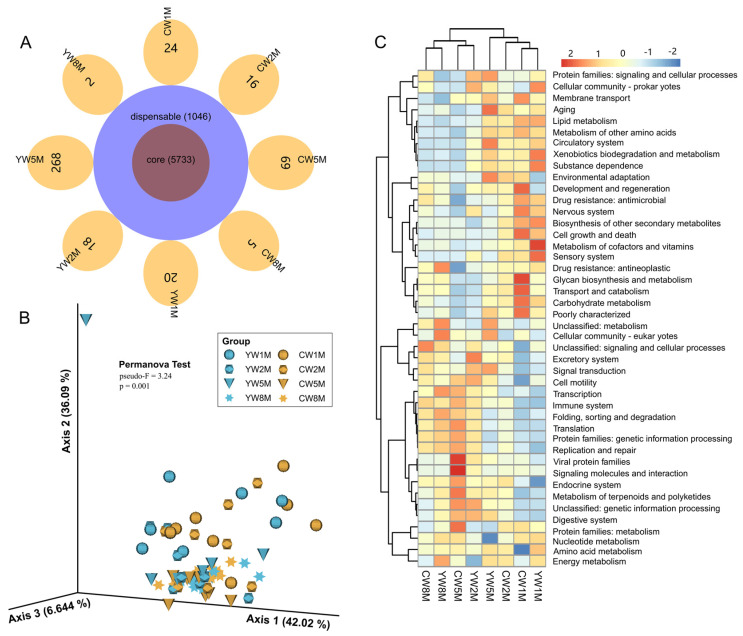
Prediction of potential functions of the fecal microbial community in weaned calves using PICRUSt2. (**A**) Veen diagram shows the sharing of microbial functional genes. (**B**) PCA plot shows microbial functional diversity across all fecal samples. (**C**) The heat map shows the differences in fecal microbial functions of yak and cattle in different months after weaning based on KEGG level 2 annotation.

**Table 1 biology-11-00031-t001:** Comparison of blood biochemical indexes of yak and cattle calves at different months after weaning.

Item	Species	Different Months				SEM	*p*-Value	
		1 M	2 M	5 M	8 M		Inter-Species	Months
GLU (mmol/L)	Yak	3.85	3.30	3.62	4.20	0.15	0.99	0.19
	Cattle	3.61	3.82	3.34	4.22	0.19		0.43
TC (mmol/L)	Yak	2.06	2.71	1.97	2.59	0.18	0.16	0.41
	Cattle	2.72	3.20	1.89	3.03	0.21		0.16
TG (mmol/L)	Yak	0.29	0.27	0.22	0.36	0.04	0.09	0.62
	Cattle	0.43	0.41	0.24	0.42	0.04		0.26
LDL-C (mmol/L)	Yak	0.44	0.46	0.47	0.51	0.06	0.21	0.98
	Cattle	0.63	0.83	0.31	0.57	0.08		0.14
HDL-C (mmol/L)	Yak	1.47	1.53	1.34	1.84	0.12	0.03	0.50
	Cattle	1.84	2.03	1.58	2.12	0.10		0.26

Note: GLU: glucose; TG: triacylglycerols; TC: total cholesterol; LDL-C: low-density lipoprotein cholesterol; HDL-C: high-density lipoprotein cholesterol; SEM: standard error of the mean. T test was used to compare the metabolic differences between weaned yak and cattle calves. One way ANOVA was used to compare the metabolic differences between weaning calves in different months.

## Data Availability

The 16S rDNA data of fecal microbiota from yak and cattle calves at different period after weaning can be freely retrieved from the NCBI Sequence Read Archive with project accession Nos. PRJNA773051 (https://www.ncbi.nlm.nih.gov/bioproject/PRJNA773051/) (accessed on 20 October 2021).

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
