# Peer review of "Fecal Microbiota Dynamics Reveal the Feasibility of Early Weaning of Yak Calves under Conventional Grazing System"

_biology, 2021, doi:10.3390/biology11010031_

Round 1

Reviewer 1 Report

In the manuscript biology-1493198 titled “Fecal microbiota dynamics revealed the feasibility of early weaning of yak calves under conventional grazing system” Jianbo Zhang and colleagues have reported that Yak (Bos grunniens) is the only bovine species adapted to the extreme ecological environment of the Qinghai‐Tibet Plateau (QTP), which provide milk, meat, transportation, fuel (yak dung), and wool for local herdsmen. However, calves are an important part of the sustainable development of the yak industry on the QTP, and the quality of calves rearing directly determines the production performance of adult animals. Under traditional grazing management, late weaning of yak calves seriously affects the improvement of their production performance. Comparative study of faecal microbiota dynamics of yak and cattle (Bos taurus) calves in different months after weaning will help to understand the changes of intestinal microbiota structure of early weaning calves. Our research will contribute to the development of appropriate strategies to regulate the gut microbiome and thus improve the growth and health of the grazing ruminants on the QTP. I have a few comments for the present manuscript.

-Thanks for the opportunity on the revision of this manuscript, 
The introduction is very well-written and this format of the journal with the simple summary is an excellent idea, maybe add more information in some lines with the relationship of health and gut microbiota 

-In the material and methods section, the authors have stated the use of OTUs, however, the accepted interpretation in our days is ASV using the DADA2 program, is possible to change OTUs for ASV.

-The way and fashion that the authors have presented their results is really good, the sample size is a good number and my main concern is again how the bacteria were selected, I know that OTUs was the first good strategy with QIIME, however with QIIME2, ASV is the good selection

-Limitations are missing in the document.

Reviewer 2 Report

Nowadays, gut microbiota gets much research attention that it deserves based on its beneficial roles in both human and animals. This study provides important information that can be used to improve the yak/cattle productivity. Furthermore, it is a well-designed, executed and written manuscript. I have the following comments:

Line 56. Due to

What was the purpose of blood draw? What research question did you want to answer using the generated data? It would be great to briefly state the purpose in the M&M.

Table 1: full names for the chemicals below the table will improve the readability

Figure 2: what do the dash horizontal lines represent?

Figure 3: Circos diagram may need more descriptions to understand it.

Line 359-60: “Our study under review presents the results that maternal fecal microbiota might be an important source of intestinal microbiota in pre‐weaning calves.” Which of your findings led to this suggestion? Because the fecal sample collection begun one month after birth.

The calves were sampled from September to May during which the type of forage and weather varied; you have highlighted the forage difference between the seasons in the discussion; would you also highlight if the change of seasons might have impacted gut microbial integrity as well?

Reviewer 3 Report

In this study, the microbiota of yak and cattle calves is compared at different times after weaning.

After reading the manuscript I have several questions and doubts for the authors which are listed below.

Minor comments.

There are undefined abbreviations the first time they are used throughout the text (e.g., LEfSe and PCA).

Line 309 and others: ... "Our study under review ..." 
Clarify this sentence. In my view it should be changed

There are grammatical errors throughout the text, e.g.

Line 346: intestinal iota

Line 354:gutmicrobiota

Table 1 and other: Define the abbreviations used in the footlegend

At what age does weaning occur in yaks and cattles?

Figure 3C: letters indicating statistical significance are not visible.

Major comments:

A comparative table of both species including their main phenotypic characteristics and differences would be helpful.  

If all the results are given for both yak and cattle, I do not understand why in the introduction and title these animals are ignored and only yak are highlighted. This should be  better claryfied  or the manuscript should be completely focused in both animals

Are cattle used as controls? In this case, why? What do the authors intend with this comparison?

As it is written, it is understood that the animals sampled during the experiment were different. Why wasn't the same animal followed throughout the experiment? Were they eutanized at every time point?

Round 2

Reviewer 1 Report

Thanks for the opportunity to revise the present manuscript, and thank you to the authors for taking into account my previous comments that may be has improved the manuscript. My congrats to the authors and the journal idea about these two abstracts is a really good idea.

Reviewer 3 Report

The revised version of the manuscript has been greatly improved. Thanks to the authors for the effort